# Alexithymia and Athletic Performance: Beneficial or Deleterious, Both Sides of the Medal? A Systematic Review

**DOI:** 10.3390/healthcare10030511

**Published:** 2022-03-11

**Authors:** Catarina Proença Lopes, Edem Allado, Mathias Poussel, Aziz Essadek, Aghilès Hamroun, Bruno Chenuel

**Affiliations:** 1Development, Adaptation and Disadvantage, Cardiorespiratory Regulations and Motor Control (EA 3450 DevAH), University of Lorraine, 54000 Nancy, France; e.allado@chru-nancy.fr (E.A.); m.poussel@chru-nancy.fr (M.P.); b.chenuel@chru-nancy.fr (B.C.); 2Center of Sports Medicine and Adapted Physical Activity, CHRU-Nancy, University of Lorraine, 54000 Nancy, France; 3INTERPSY (EA 4432), University of Lorraine, 54000 Nancy, France; aziz.essadek@univ-lorraine.fr; 4Department of Public Health, Epidemiology, Health Economics and Prevention, Regional and University Hospital Center of Lille, Lille University, 59000 Lille, France; aghiles.hamroun@chru-lille.fr

**Keywords:** alexithymia, sport psychology, emotion, performance, athlete

## Abstract

Background: Numerous studies have been published on alexithymia among athletes in the last decades. The objective, here, is to provide a critical review on alexithymia in sport and identify elements demonstrating that alexithymic athletes can attain a competitive advantage. Methods: The Center for Reviews and Dissemination guidelines were used. The Preferred Reporting Items for Systematic Review and Meta-Analysis (PRISMA) guidelines served as the template for reporting the present systematic review. We searched PubMed, Embase, Science Direct, and PsycINFO, without language or date restrictions. Results: Within 72 eligible studies, 23 articles fulfilling the selection criteria were included in the review. Alexithymia is associated with various pathologies and considered to be counter-performing. However, despite considerable suspicion of an advantageous performance effect of alexithymia, there is a lack of data to quantify this effect. Studies identified are heterogeneous (different scales of measurement of alexithymia used or outcomes, different sports), that do not allow us to conclude on an observed causal relationship, because the studies are mostly observational. Conclusion: This systematic review opens a new search field on alexithymia, as possibly promoting performance.

## 1. Introduction

Following the very first description by J. Nemiah and P.E. Sifneos, on the eighth European Conference on psychosomatic research, in 1970 [1], alexithymia is nowadays recognized as a personality trait, common in the general population [2]. The prevalence varies between 8% and 23% [3]. Alexithymia refers to a deficient ability to recognize and describe emotions, use of concrete speech and thoughts related to external events, and a paucity of fantasy life [4]. Further, alexithymia affects the recognition of one’s internal state [5,6].

Initially considered as a mode of pathognomonic functioning of patients suffering psychosomatic component diseases, it would be shown that there was a high comorbidity between alexithymia and addictive behaviours (alcoholism, substance abuse, eating disorders: anorexia and bulimia) [7]. Therefore, alexithymia was firstly associated with a number of psychosomatic illnesses, but could also represent vulnerability factors, favorising the development of psychiatric disorders in adulthood (depression, anxiety, post-traumatic stress disorder, etc.) [7,8,9,10,11,12]. Alexithymia is not exclusively observed in health condition but also in other domains, such as sport. This was more recently highlighted in the field of performance research, playing a key role in risk-taking behaviours [13,14]. High-risk sports have typically been investigated within a sensation-seeking framework [15]. A person with alexithymia may find it easier to identify emotions in high-risk settings, where emotions, such as fear, may be more easily identified than in other domains. Therefore, they first experience fear and then the associated reduction after completing the activity [14,16,17,18]. Studying extreme mountaineers, ocean rowers and skydivers, Woodman et al. showed that alexithymic individuals derive adapted emotion regulation benefits as a result of engaging with high-risk sports [13,14,17].

Emotion regulation requires an individual to initiate, maintain and modulate the intensity and duration of their emotions. Emotional management is important in high performance sport. Anxiety is the most studied emotion in sport psychology. Several theories have developed around the concept of anxiety and, notably, the conceptualization in two dimensions, independent of each other: cognitive (conscious subjective feelings of apprehension and tension, caused by pessimistic expectations of success or negative self-assessments) and somatic anxiety (which refers to the physiological dimension and, in particular, to the notion of activation) [19]. Anxiety influences the performance, both positively and negatively [20]. The benefits of regulating emotion in everyday life also take place in the ability to deal with close relationships [18].

More generally, in sports, and not only in the peculiar high-risk domain, the stressful nature of the competitive environment could lead to the intriguing possibility that alexithymic individuals may be drawn to competitive settings and then, alexithymia could be a master trump for performance [21]. Alexithymic athletes would be more able to adapt effective emotion regulation in these competitive conditions, as they do in high-risk domains.

The hypothesis that performance of alexithymic individuals might be less affected by high levels of anxiety is attractive. Then, the ability to regulate emotions successfully is regarded by many athletic trainers as an important psychological skill in athletes and they firmly intend to develop it [22,23,24].

This systematic review aims to summarize the current evidence on the association between alexithymia and sports practice. Studies will be classified into four categories: anxiety and depression, overtraining (burnout), addiction and risky sports behaviour and, finally, alexithymia and sport.

## 2. Materials and Methods

### 2.1. Design

The Center for Reviews and Dissemination guidelines were used for the methodology of this review. The Preferred Reporting Items for Systematic Review and Meta-Analysis (PRISMA) guidelines served as the template for reporting the present review [25] (Figure 1).

### 2.2. Search Strategy and Selection Criteria

We searched Embase, Science Direct, PsycINFO and through PubMed without language or date restriction. The search strategy applied to Embase was adapted, in order to fit with other databases (Table 1). To supplement these database searches, references of all relevant studies were also screened to identify additional potential data sources. Database inception to 31 January 2021, then updated until 16 January 2022.

We considered observational (case-control studies, prospective, and retrospective cohorts) and interventional studies. We excluded letters, commentaries, editorials, and studies with no data available after two unsuccessful requests sent to the corresponding author. For studies published in more than one report (duplicates), we considered the most comprehensive study that reported the largest sample. Selection process was carried out in three stages (Figure 1); papers were first reviewed by title, then by abstract, and finally by full text [26,27]. Two reviewers independently appraised identified literature. At each stage, articles were excluded if they did not meet the following inclusion criteria: be a study about impact of alexithymia on sport practitioners.

The methodological quality of the included studies was assessed using the CASP tool (Critical Appraisal Skills Programme). Studies were categorized as follows (Table A1 in Appendix A):Red color for a negative response to the criteriaYellow color for an uncertain response to the criteriaGreen color for a positive response to the criteria

### 2.3. Data Extraction and Management

The data extraction and synthesis of study results was conducted by peer review. This process involved two authors independently appraising papers: the lead author and an academic from an affiliated institution (the second author) [28] then reaching a consensus over the final study appraisal through debate.

Using a pretested data-extraction form, relevant information was extracted, including first author, publication year, country of recruitment, study design, participant characteristics (number/size, age, and gender), type of sport and level of practice, alexithymia assessment scale, outcomes and, finally, main results (Table 2).

## 3. Results

### 3.1. Review Process

We initially identified 2916 records; after duplicate exclusion, 2793 remained. We screened their titles and abstracts and excluded 2721 irrelevant records. Agreement between investigators on abstract selection was κ = 0.98.

We analyzed the full texts of the remaining 72 papers for eligibility, of which 50 were excluded. Additional records identified through database researching after the update in 2022 identified 35 more records, but only 1 text was included.

Finally, as presented in Table 1, a total of 23 full texts were retained. The inter-rater agreement between investigators was κ = 0.93 for final study inclusion.

### 3.2. Characteristics of Included Studies

In all, 23 studies including a total of 5648 participants, from 10 countries, were included in this study. One randomized study, one prospective study, eighteen cross-sectional studies, and three longitudinal studies.

### 3.3. Review of the Studied Alexithymia and Athletic Performance

Studies were classified into four categories, according to the different theme on alexithymia investigated in sport: anxiety and depression, overtraining (burnout), addiction and risky sports behaviour, alexithymia and sport (Table A1 in Appendix A).

### 3.4. Anxiety and Depression (Five Studies)

The most studied emotion in sports psychology, a significant association was observed between anxiety and alexithymia. Alexithymic athletes would be more anxious than non-alexithymic athletes [13,14,17]. Alexithymia would be a potential moderator of anxiety fluctuations, in high-risk sport or high-risk domains, generally. For example, Alexithymia moderated the pre- to post-jump fluctuation, such that, only alexithymic skydivers’ anxiety diminished as a consequence of performing a skydive, compared to non-alexithymic skydivers.

The data in the present study suggest that the alexithymic skydivers may purposefully expose themselves to anxiety-provoking situations [14].

Furthermore, one study suggested that levels of depression and alexithymia seem to be predictors of competitive anxiety. Thus, the higher the levels of depression and alexithymia are, the more likely subjects would be anxious when approaching a competition [29].

To our knowledge, the link between alexithymia and depression in sports has been little studied, in contrast to the relationship between alexithymia and anxiety or depression in the general population. Results appeared different, depending on samples and conditions. In the study by Aston et al. [30], for retired and active hockey players, greater alexithymia was associated with greater depressive symptoms. Medina-Porqueres et al. [31] showed that seniors who are physically active have lower results in alexithymia and depression scores than those who are not physically active.

### 3.5. Overtraining (Burnout) (Four Studies)

The concept of burnout is also observed in alexithymic athletes. They would over-adapt to environments without being aware of their own internal sensations (physical and emotional exhaustion), improving a decrease in motivation for competitions and leading to burnout. A positive correlation was observed between alexithymic tendencies and burnout (r = 46, *p* < 0.01, 95% CI (31,0.59)); [32,33]. The risk of burnout in athletes is related to the intensity of training. The studies by Zekioglu et al. [34] and Allegre et al. [35], show that alexithymic athletes train more intensely.

### 3.6. Addiction and Risky Sports (Ten Studies)

Addiction to individual sports, repetitive exercises with an emotional detachment of the task to be performed is noted in alexithymic athletes [36]. Some addiction to risky sports can also be observed in alexithymic sportsmen. This is explained by the short-term emotional benefits (especially in terms of anxiety) brought by this sport, causing a continuing search for regulation through its practice [37,38]. For Van Landeghem et al. [39], alexithymia would play a role in maintaining dependence on exercise, while for Andres et al. [40], it could lead to alcohol with sport students. However, in addition to the explanation of apparent common risk factors, towards even different addictions (physical exercise and alcohol), the association with the two different conditions could also be structured with different psychopathological and pathophysiological modalities. Gori et al. [41] showed the positive relationship between alexithymia, exercise addiction, body image and self-esteem. Indeed, persons with alexithymia tend to have difficulty in managing their affect. This may lead to an excessive focus on physical components and body image distortions, so as to avoid contact with the emotional experiences. Therefore, for the authors, engaging in addictive behavior may become a dysfunctional strategy to cope with painful emotion.

The high-risk environment is at the center of research involving alexithymia in sport. Theories suggest that this environment would provide the benefits of regulating emotions and, thus, experience and mastering anxiety, which can be particularly beneficial [14,37,42,43,44].

### 3.7. Alexithymia and Sport (Four Studies)

Except for risky sports, there are few data exploring the association between alexithymia and sports, with highly heterogenous conclusions. Indeed, Demir. H [45] assessed that athletes with disabilities have higher scores in Alexithymia, compared to non-sportive individuals with disabilities. In the study of Jodat et al. [46], sports students were less alexithymic than non-sport students. Finally, a study of winter swimmers (people who take a dip in a hole in ice-topped natural waters regularly throughout the winter) showed no significant difference between winter swimmers and controls (swimmers who does not participate in winter swimming) in Toronto Alexithymia Scale 20 variables [47].

Another recurring element in the research is the negative view of alexithymia, which would be deleterious to any performance. Athletes are generally able to flexibly use a range of strategies to regulate emotions in response to changing contexts and psychological demands [48]. Being alexithymic in sports is associated with lower levels of tenacity. But in another way, perfectionism is associated with high rates of alexithymia [48]. This personality trait is defined as a commitment to extremely high standards and self-critical evaluative tendencies [49]. It is a prevalent characteristic in athletes. It is a multidimensional characteristic and only some dimensions of perfectionism are clearly maladaptive, whereas perfectionistic strivings may form part of a healthy striving for excellence [50].

## 4. Discussion

The aim of this systematic review was to summarize the current evidence about the association between alexithymia and sports practice. Our findings suggest that alexithymia is poorly studied in the field of sport. Alexithymia is mostly associated with disorders, such as anxiety, depression, overtraining (burnout), addiction and risky sports behaviour. Nevertheless, it appears that some characteristics of alexithymic athletes could be beneficial to performance.

The few data in high-level sports seem to emphasize a link between alexithymia and burnout. Indeed, according to Amemiya and Sakairi, [32] “athletes are thus unable to appropriately manage their psychosocial (and sometimes physical) problems or to obtain support from others, and consequently, they experience”. Therefore, they experience a long-term exposure to several stressors. They are prone to burnout that would lead to interpersonal and emotional exhaustion and a lack of motivation [51]. Alexithymic athletes fall into an over-adapted state without reflecting on their condition or internal experience. Therefore, they probably do not experience positive emotions, which lead to stronger motivation for one’s activities [52]. Burnout is well present in sport, as shown by Gerber et al. [53], in their study of an elite sports population, where about one in ten young elite athletes (12%) reported symptoms of burnout or depression (9%), and more recently, athletes with high burnout symptoms tend to give up sport [54]. It would be interesting to continue to investigate the link between alexithymia and burnout, to be able to better support athletes and prevent burnout.

The intensity of training is one of the parameters that can induce mental fatigue and would influence the mood of athletes [55]. Indeed, it is known that, in young athletes, high-intensity training can cause emotional problems, injuries, anorexia, and a disruption of the relationship with others [56]. The studies of Zekioglu et al. [34] and Allegre et al. [35], underline the association between the athlete’s alexithymia and the practice of more intense training, which would be the origin of the burnout. Alexithymic athletes would be more at risk to develop burnout, so we can question the notion of limits in alexithymic athletes. Amemiya and Sakairi [32] explain this by the lack of awareness of their state of physical and mental fatigue. It can, therefore, be assumed that these athletes would be able to withstand loads and intensities of higher training sessions than non-alexithymic athletes. This may, therefore, be an advantage, if a specific prevention work is implemented in order to avoid burnout.

For Woodman and Hardy [57], the deterioration in athletes’ performance could be explained to 10% by an increase in cognitive anxiety. Cognitive anxiety is defined as “conscious subjective feelings of apprehension and tension, caused by pessimistic expectations of success or negative self-assessments” [19]. It influences performance [20] and can also influence the acquisition of a new gesture, requiring more time for the athlete to appropriate it [58,59]. The prevalence of anxiety in athletes differs from one study to another, from type of sport, their gender or country [60,61,62,63,64], and can range, for example, from 5.9% to 34% (the prevalence of anxiety in the global population is rather close to 3.8% [65]). This idea was highlighted in a study by Tim Woodman (2001), among athletes in high-risk sports. Indeed, they compared the environment of high-risk sports to that of competitions, by making a parallel with the threat on the ego, if the athlete loses. Therefore, the investment of the athlete is at maximum, in order to bring him/her the vital benefit of winning. This showed that pre-competition anxiety increases comparably to the anxiety of high-risk sports athletes. Alexithymic athletes would, therefore, have an advantage over non-alexithymic athletes, by experiencing and overcoming the anxiety generated by the competition [66,67]. According to Woodman et al. [66], the difficulty of regulating emotions, although problematic on a daily basis, could be advantageous in the high-pressure field of competitions. In fact, it allows alexithymic athletes to experience, and then master, an emotional clarity in everyday life. The mastery of anxiety in these settings then facilitates a relative sense of well-being after sport. Because alexithymics have emotional control difficulties, such experience and mastery of anxiety may be particularly beneficial. For other alexithymic athletes, they may be attracted to anxiety-inducing physical sensations, such as adrenaline, which was seen for high-risk domains. As for high-risk environment, we suggest that competitive conditions would provide the benefits of regulating emotions and, thus, experience and mastering anxiety, which can be particularly beneficial to achieve performance. This hypothesis was also formulated by Hardy et al. [21], following their qualitative psychosocial study on elite English athletes.

Howe et al. [68], demonstrated in a study of marathoners, a significant increase in cortisol concentration in athletes with a high emotional intelligence trait. Emotional intelligence being considered as negatively related with alexithymia, we can suppose that this personality trait would increase the stress awareness of marathoners, induced by the rise in cortisol before the race. These results would allow us to make the following hypotheses, that alexithymia would be a protective resource when facing stress situations. However, despite the attractive hypothesis of the beneficial effect of alexithymia on performance, there is a lack of data and proof supporting it.

The majority of articles on alexithymia in sports are about addiction disorders or identified in at-risk sports. Namely, in the general population, alexithymia would significantly predict preferences for aggressive and illegal behavior, at-risk sexual behaviour [69]. Furthermore, a high prevalence of alexithymic people is observed among alcoholics [70], drug-addicted patients [7] and eating disorders, such as bulimia [7,10,11]. Alexithymia and the presence of more aggressive and irresponsible behaviour in the school/work environment is also supported in the study by Panno et al. [71].

In the field of sports, a peculiar attraction for risky sports has been noted among alexithymic athletes, but also an addiction to individual sports, repetitive exercises with an emotional detachment from the task to be performed [35]. These studies have all identified and focused their work on aspects deleterious to the health of the athlete. While even stress management is noted for risky sports activities, attention is paid to the dangerous nature of this sport. As in the general population, alexithymia in sports is associated with pathology and handicap, therefore, against performance. However, the management of delicate and anxiogenic or stressful situations, resistance to high loads and training intensities, with a high tolerance for pain, are perceived as qualities sought in high-level sport. All these characteristics were noted in this literature review but not analyzed/studied in alexithymic sports men and women. However, it would be interesting, given these elements, to explore alexithymia as a possible factor facilitating performance.

### Study Limitations and Future Directions

Our results highlight the relative lack of research on alexithymia in sports and, more particularly, by questioning the existence of a link with performance. To our knowledge, it is the first systematic review to put forward characteristics in alexithymic athletes that hypothesize a link to performance: doing more intense training, being less sensitive to the effects of anxiety in the high-pressure field of competitions, and being a perfectionist. This does not exclude the possibility for alexithymic athletes to also encounter deleterious properties of its characteristics, which underlines the need for appropriate support by the coach and the staff with knowledge of the risks for alexithymic athletes.

However, this study presented some limitations. The studies identified are heterogeneous: different population, different sports, athletes of different levels, different scales of measurement of alexithymia used or outcomes, that do not allow us to conclude on an observed causal relationship because the studies are mostly observational.

After analysis, we note the importance of absence or low presence of limits and, especially, an attraction for the thrills, the adrenaline noted in studies on sports at risk. In high-risk domains, alexithymic prevalence is markedly higher than in the general population, reaching 30% in climbing, scuba-diving, skydiving, sea-rowing, rowing and mountaineering [13,14,17], varying from 8% to 20% among the general population [2]. It would be interesting to explore, outside of extreme sports, the relationship between the amount of sports training and level of alexithymia, evaluate the prevalence of alexithymia in different types of sports (confrontation, collective, individual) and to study the relationship with the amount of training and the practice level (recreational versus competitive practice) in individuals who are officially licensed in a sport club.

Another research perspective could explore the causes of alexithymia development in athletes. Are they strongly alexithymic before turning to sport or is the practice of sport at a certain level provoking alexithymia? In recent research, the family link on the development of alexithymia has been highlighted. The total levels of alexithymia and their sub-dimensions of families, whose children were involved in sports, were significantly higher than children who were not involved in sports. The level of alexithymia and social competence of families can, thus, affect the orientation of their children to sports [72]. The type of education can also influence the development of alexithymia. The gender difference would be marked mainly in the verbalization of emotions and not in the identification explained by a conditioning of the social role of man, in connection with a bias of social desirability, when passing the Toronto Alexithymia Scale questionnaire (TAS 20) [73]; different parental attitudes, according to the sex of the child and cultures [74]. It would be interesting to see the prevalence of alexithymia among athletes stratified by gender. Furthermore, perhaps it is possible that sports practice attracts people with alexithymia in certain types of sports (confrontation, team sports, individual sports, etc.). For example, alexithymic people would be more oriented towards individual sports, as alexithymia is associated with poor interpersonal relationships or that alexithymia is deleterious in team sports. We have seen that alexithymia is associated with a series of negative outcomes, so it will be interesting to explore the possibility that sports practice attracts people with alexithymia in certain circumstances (disease, divorce, etc.).

These new research perspectives would allow the sports community to offer adapted support to alexithymic athletes, in order to allow them to develop their potential, while allowing them to be in good mental and physical health.

## 5. Conclusions

Alexithymia is associated with various pathologies and considered to lead to under-performance. This present systematic review has highlighted the lack of papers about a link between alexithymia and sports performance. We suggest an innovative field of research on alexithymia as possibly promoting performance.

## Figures and Tables

**Figure 1 healthcare-10-00511-f001:**
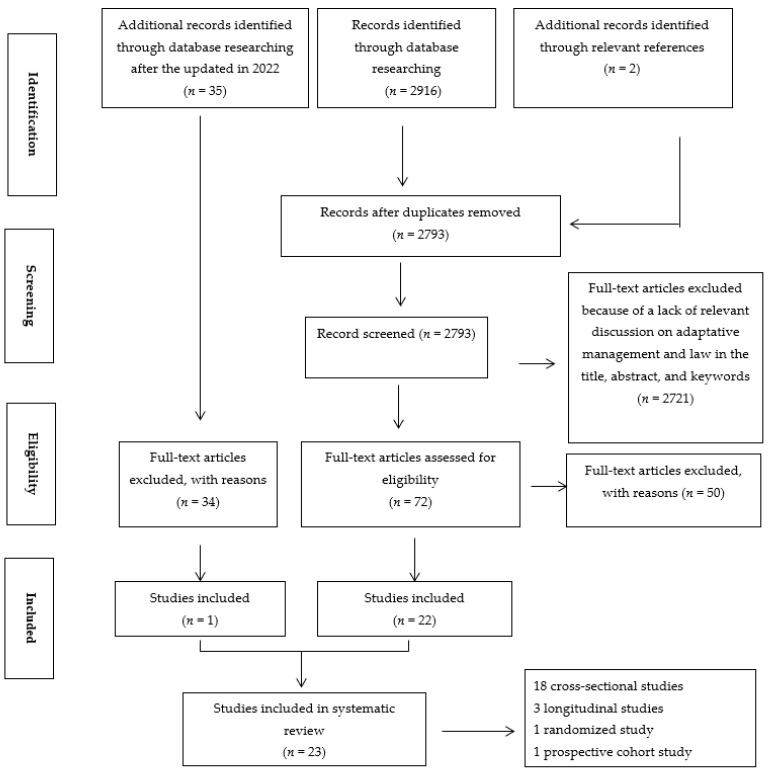
Flowchart outlining the protocol adopted in this systematic review based on the Preferred Reporting Items for Systematic Reviews and Meta-Analyses (PRISMA).

**Table 1 healthcare-10-00511-t001:** Search Strategy.

Criterion	Detail
search terms	alexithymia, affective symtom, emotional disturbances, exercise, physical activity, sport, trainings, athletics, aerobic
Language	no restrictions
Timeframe	no restrictions
Database	embase, science direct, PsycINFO, PubMed
Inclusion criteria	observational and interventional studies
Exclusion criteria	letters, commentaries, editorials, and studies with no data available after two unsuccessful requests sent to the corresponding author
Initial search results	2953
Included in review	23

**Table 2 healthcare-10-00511-t002:** Reviewed studies characteristics.

	Authors	Study Design	Participants Characteristics	Type of Sports, Level	Alexithymia Assesment Scale	Outcomes	Results
Anxiety & depression	Woodman & al, UK (2008)	longitudinal	N: 111, 100% Women; Mage: 23 years	skydiving (experienced)	TAS-20	Anxiety, Sensation seeking	Higher anxiety in alexithymics
	Arnaud & al, France (2012)	cross-sectional	N:150; 77% Men, Age 10–62	tennis (high-level)	TAS-20	Locus of control, Anxiety, Depression	Alexithymia predict competitive anxiety
	Barlow & al, UK (2015)	cross-sectional	N: 1358, 85% Men, Mage: 34 years	variety of high-risk sports (experienced)	TAS-20	Sensation Seeking, Emotion Regulation, Risk taking, Anhedonia	Alexithymia mediated by deliberate risk taking and precautionary behaviors
	Medina-Porqueres & al, Spain (2016)	cross-sectional	N:27, 67% Women, Mage: 64 years	not described	TAS-20	Quality of life, Depression	Lower alexithymia in physically active elderly people
	Aston & al, USA (2020)	cross-sectional	N: 409, 100% Men, Mage: 32 years	hockey (high-level)	The Brief Form Normative Male Alexithymia Scale	Depression, Anxiety, Perceived social support	Association with greater depressive symptoms
Overtraining (burnout)	Allegre & al, France (2007)	cross-sectional	N:20, 100% Men, Mage: 20 years	swimming (experienced/high level)	ALCESTE	-	Link with severe constraints and hardships of practice
	Zekioglu & al, Turkey (2014)	cross-sectional	N: 95, 77% Men, Mage: 21 years	not described	TAS-20	polymerase chain reaction method, weekly training hours	No significant relationship with training intensity
	Amemiya & al, Japan (2015)	cross-sectional	N: 353, 63% Men, Mage: 21 years	contactless sport, group sport (baseball, etc.) individual sport (soft tennis etc.) (high level)	SAS	Mindfulness, Burnout	Mindfulness affected athletes’ burnout by decreasing alexithymia
	Amemiya & al, Japan (2018)	longitudinal	N: 125, 59% Men, Mage: 20 years	tennis, soft tennis, dance, football, softball, and soccer (high level)	SAS	Mindfulness, Burnout, Psychological Performance	Mindfulness reduced alexithymic tendencies
Addiction and risky sports behavior	Cazenave & al, France (2007)	cross-sectional	N: 180, 100% Women, Mage: 26 years	non-risk sports, risk-taking sports (experienced/leisure time)	TAS-20	Sensation Seeking, Sex Role, Impulsiveness, Risk & Excitement	Link with risk-taking behaviors in professional women’s
	Lafollie & Le Scanff, C, France (2007)	cross-sectional	N: 274, 100% Men, Mage: 26 years	mountaineers snowboarders, mountain biking, gymnasts, basket (experienced/high level)	TAS-20	risk & activation, Anxiety, Sensation-Seeking, Risk & Excitement	Looking for physical sensations, but not necessarily by adopting of dangerous lines
	Woodman & al, UK (2009)	cross-sectional	N:87, 64% Men, Mage = 30 years	Skydivers (experienced)	TAS-20	Anxiety, Heart rate, Sensation Seeking	Higher anxiety than their nonalexithymic counterparts.
	Woodman & al, UK (2010)	longitudinal	N: 44, 95% Men, Mage: 33 years	rowers and mountaineers (experienced)	TAS-20	agency, emotion regulation, sensation seeking	Link with participants of prolonged engagement high-risk sports
	Andres & al, France (2014)	cross-sectional	N: 434, 46% Women, Mage: 20 years	tennis, soccer, athletics, swimming, judo and various (experienced)	TAS-20	Alcohol, Peer Attachment, Personality	Effect of low conscientiousness and alexithymia between maternal insecure attachment and alcohol use
	Manfredi & al, Italy (2015)	cross-sectional	N: 137, 59% Men, Mage: 32 years	Sport centers (leisure time)	TAS-20	Exercise Dependence	Correlation with exercise addiction
	Bonnet & al, France (2017)	cross-sectional	N:131, 89% Men, Mage 40 years	scuba diving (experienced)	TAS-20	Emotionality, Risk-Taking, Personality	Factor contributing to short-term risk taking
	Calogero & al, Italy (2017)	cross-sectional	N: 200, 63% Women, Mage 25 years	not described (experienced/no sport)	TAS-20	Body Uneasiness	Link with exaggerated practice of sport
	Van Landeghem & al, Canada (2019)	cross-sectional	N: 600, 66% women, Mage: 19 years	not described	BVAQ	Exercise dependence, Eating Disorder, Health, Interpersonal Reactivity, Attention checks	Link between Alexithymia and Exercise dependance
	Gori & al, Italy (2021)	cross-sectional	N: 288, 72% women, Mage: 28 years	not described	TAS-20	Exercise addiction, body image, self esteem	association between alexithymia, exercise addiction and body image
Alexithymia & Sport	Lindeman & al, Finland (2002)	prospective	N: 25, 72% Women, Mage: 55 years	Winter swimming (leisure time)	TAS-20	anxiety, obsessionality, depression, somatic anxiety, and hysteria	No major differences between winter swimmers and controls
	Jodat & al, Iran (2015)	randomized	N: 200, 100% Men, Mage: 16 years	Sport club (No sport/athletes)	FTAS-20	-	Athletics students are less alexithymic
	Demir & al, Turkey (2018)	cross-sectional	N: 339, 73% Men, Mage: 25 years	Sport centers (leisure time)	TAS-20	Social comparison	Disabled athletes had higher alexithymia scores compared to disabled non-athletes
	Kucharski & al, Canada (2018)	longitudinal	N: 61, 54% Women, Mage: 20 years	Team sports (experienced)	TAS-20	coping strategies, stress, self regulation, personality	Association with perfectionism and lower levels of toughness

TAS-20: The Toronto Alexithymia Scale, Alceste: method of discourse analysis, SAS: Sport Alexithymia Scale, BVAQ: Bermond Vorst Alexithymia Questionnaire, FTAS-20 is a Farsi version of TAS-20 validated for the Iranian population.

## Data Availability

Not applicable.

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
