# Peer review of "Alexithymia and Athletic Performance: Beneficial or Deleterious, Both Sides of the Medal? A Systematic Review"

_healthcare, 2022, doi:10.3390/healthcare10030511_

Round 1

Reviewer 1 Report

The aim of this review is to identify evidence that illustrates the positive effect of alexithymia in a competitive sport situation. After a rigorous search, the authors included 23 papers in their study. Although this study is very interesting, the introduction needs to be expanded and the discussion reworked to better understand the authors' assumptions about the link between alexithymia and performance. Furthermore, the positive link between alexithymia and performance is purely hypothetical and the data set provided by the authors is contradictory to their hypotheses.

In the attached document, I offer the authors my comments. 

Reviewer 2 Report

I would restructure the article more than as a systematic review on a specific question (i.e. is alexithymia associated with better or worse performance?) Towards a scoping review on sport / exercise / performance and alexithymia

Aim of the study

The discussion starts with the sentence  “The aim of this systematic review was to summarize the current evidence about the association between alexithymia and sports practice”

which is interesting and even partially well discussed but is different from when announced in the introduction. The introduction had previously discussed a study published in a book, difficult to find, which would have found a positive association between the presence of alexithymia and good performance in athletes. So, it claimed that the hypothesis that “performance of alexithymic individuals might be less affected by high level of anxiety is attractive. Then, the ability to regulate emotions successfully is regarded by many athletic trainers as an important psychological skill in athletes and they firmly intend to develop it [15-17]. This systematic review aims to summarize the current evidence on the association 50 between alexithymia and sports performance" .

Now I believe that even if the study started with the hypothesis of verifying the association between sports performance and alexithymia, the article does not discuss this hypothesis at all, perhaps because it did not find enough material. In my opinion, even if a study had found this type of association, the issue is so complex that the result should be contextualized to the type of sport and the type of athletes. Even the type of coach and his style can in some cases select athletes with alexithymia, so a generalization in this field is in my opinion difficult to support and demonstrate. But since, in reality, the article does not discuss this hypothesis, I would propose to modify the introduction, talk about scoping review on sport e alexithymia, to emphasize even more (in part the authors already do) the importance of the context variables that make generalizations difficult and ending with more precise indications on the future direction of research.

Minor points

“Moreover, results appeared controversial. In the study by 129 Aston et al., [23] for retired and active hockey players, greater alexithymia was associated 130 with greater depressive symptoms. In contrast, Medina-Porqueres et al.,[24], have shown 131 that seniors who are physically active have lower results in alexithymia and depression 132 scores than those who are not physically active”

These two results are not in contradiction, simply the focus is on totally different samples and conditions. The fact that in former athletes alexithymia is associated with depression could be due to the fact that the presence of alexithymia is associated with a difference in adaptation after the cessation of competitive sport, this type of condition has nothing to do with the elderly in the civil society in which many reasonable different hypotheses can be produced to explain the association between alexithymia and physical exercise (not professional)

What has been discussed about overtraining and alexithymia seems reasonable

“alexithymia would play a role in maintaining dependence on exercise while for Andres et 148 al., [33], it could lead to alcohol with sportstudents”

perhaps it should be specified that in addition to the explanation of an apparent common risk factors towards even different addictions (physical exercise and alcohol) the association with the two different conditions could also be structured with different pathophysiological and psychopathological modalities in the two conditions

“In contrast, a study in Amyotrophic 149 Lateral Sclerosis patients showed no link between exercise addiction and alexithymia 150 (Roy-Bellina S. et al.,[34]”

however the condition of suffering from amyotorphic sclerosis is so specific that this condition can hardly be compared with that of people who do not suffer from it also with regard to the factors associated with alexithymia

“Emotional intelligence being considered as the opposite of alexithymia”

This statement is simplistic, the two dimensions concern different aspects, although it is hypothesized that high alexithymia and low social intelligence can be associated with it, it is not certain that there are no exceptions, moreover the theme does not seem sufficiently studied to support such a categorical statement

“Our results highlight the relative lack of research on alexithymia in sports and more 253 particularly on the fact that it could facilitate performance”

In the whole article, I would consider the aspect of the association between alexithymia and sports performance in a very marginal way

“To our knowledge it is the first 254 systematic review to put forward characteristics in alexithymic athletes that could be ben- 255 eficial to performance”

It does not add anything on this aspect which is so complex given the current knowledge that it is even risky to deal with

“Another research perspective could explore the causes of alexithymia development in athletes”

This is interesting, that is, alexithymia is associated with a series of negative outcomes and it is possible that sports practice attracts people with alexithymia in certain circumstances and in certain types of sports, it is also possible that sports practice may improve under certain conditions, and in certain others to worsen the alexithymia, these should be the points to be explored

Round 2

Reviewer 1 Report

I thank the authors for the corrections made to this paper. The introduction seems to me to be more fluid. However, there are still points to be explored. Also, I strongly advise the authors to read the chapter "Alexithymia" in “the Routledge International Encyclopedia of Sport and Exercise Psychology” (2020), or even to reference it.

My comments are attached.
